# Update on Findings about Sudden Sensorineural Hearing Loss and Insight into Its Pathogenesis

**DOI:** 10.3390/jcm11216387

**Published:** 2022-10-28

**Authors:** Satoshi Yamada, Junya Kita, Daichi Shinmura, Yuki Nakamura, Sosuke Sahara, Kiyoshi Misawa, Hiroshi Nakanishi

**Affiliations:** 1Department of Otorhinolaryngology/Head & Neck Surgery, Hamamatsu University School of Medicine, Hamamatsu 431-3192, Japan; 2Department of Otorhinolaryngology, Numazu City Hospital, Numazu 410-0302, Japan

**Keywords:** sudden sensorineural hearing loss, COVID-19, thrombosis, cochlear inflammation, viral infection, SARS-CoV-2

## Abstract

Sudden sensorineural hearing loss (SSNHL) is routinely encountered and is one of the most common emergent diseases in otolaryngology clinics. However, the etiology of SSNHL remains unclear. Due to the inaccessibility of the living human inner ear for biopsy, studies investigating the etiology of SSNHL have been performed by analyzing data obtained from examinations using peripheral blood or imaging. We updated the findings obtained from serological, magnetic resonance imaging, genetic, and viral examinations to reveal the etiology of SSNHL. Regarding viral examination, we focused on sensorineural hearing loss associated with coronavirus disease (COVID-19) because the number of correlated reports has been increasing after the outbreak. The updated findings revealed the following three possible mechanisms underlying the development of SSNHL: thrombosis and resulting vascular obstruction in the cochlea, asymptomatic viral infection and resulting damage to the cochlea, and cochlear inflammation and resulting damage to the cochlea. Thrombosis and viral infection are predominant, and cochlear inflammation can be secondarily induced through viral infection or even thrombosis. The findings about sensorineural hearing loss associated with COVID-19 supported the possibility that asymptomatic viral infection is one of the etiologies of SSNHL, and the virus can infect inner ear tissues and directly damage them.

## 1. Introduction

Sudden sensorineural hearing loss (SSNHL) is defined as the rapid onset of subjective symptoms of hearing impairment in one or both ears that are sensorineural, occur within 72 h, and are accompanied by hearing loss of 30 dB or more in at least three contiguous audiometric frequencies [1]. Hearing loss can develop either immediately or in a few hours, and is typically unilateral, with bilateral involvement of less than 2% [2]. Patients frequently complain of tinnitus and aural fullness in the affected ear as accompanying symptoms, with some patients experiencing vertigo. The incidence of SSNHL has been reported to range from 5/100,000 to 150/100,000, and it predominantly occurs at the age of 41–55 years [3,4]. The treatment for SSNHL is controversial; however, the systemic administration of corticosteroids is the most commonly used initial therapy [5]. After treatment, complete recovery has been observed in approximately 40% of patients with SSNHL [6].

Despite extensive investigation, the etiology of SSNHL cannot be determined in approximately 90% of patients and is referred to as idiopathic sudden sensorineural hearing loss (ISSNHL) [7]. However, three possible mechanisms underlying the development of SSNHL have been proposed: vascular compromise, chronic inflammation, and viral infection [8,9,10]. Due to the inaccessibility of the living human inner ear for biopsy and pathological analysis, studies investigating the etiology of SSNHL have been performed by analyzing data obtained from examinations using peripheral blood or imaging. In this review article, we updated the findings obtained from serological, magnetic resonance imaging (MRI), genetic, and viral examinations to shed more light on the etiology of SSNHL. Reports regarding sensorineural hearing loss caused by viral infection were rare before the outbreak of coronavirus disease (COVID-19); however, sensorineural hearing loss caused by severe acute respiratory syndrome coronavirus 2 (SARS-CoV-2) infection has been increasing [11,12]. In the section of viral examination, we focused on sensorineural hearing loss associated with COVID-19 to investigate the pathogenesis and to reveal the association between SSHNL and viral infection.

The purpose of this review was to understand the etiology of SSNHL and elucidate the unknown etiology of ISSNHL. For this reason, SSNHL caused by the autoimmune disease, noise exposure, physical damage, and acoustic tumor were not included.

## 2. Materials and Methods

### 2.1. Database Search

We obtained relevant literature published between 2013 and 2022 from PubMed and Embase using the following medical terms for the review article: hearing loss, SSNHL, neutrophil, monocyte, platelet, lipid for the sections of serological examinations; hearing loss, SSNHL, MRI for the section of MRI examinations; hearing loss, SSNHL, polymorphism, mutation, microRNA for the section of genetic examinations; and hearing loss, SSNHL, COVID-19, SARS-CoV-2, coronavirus, viral infection for the section of viral examinations. These words were combined with “and/or” for each section and searched. The retrieval scheme was based on a combination of medical subject headings (MeSH) terms and free words. Two reviewers (S.Y. and H.N.) independently assessed the eligibility of the studies and extracted the data.

### 2.2. Inclusion and Exclusion Criteria

We included articles from clinical series and reviews published in relevant journals with consistent and well-defined materials and methods. However, articles that were presented as abstracts, experts, letters, or comments with different themes of interest and repetitions were excluded. Articles not available in full-text or those that were not written in the English language were also excluded. Additional papers published before 2013 were identified through the references in the published literature. Full-text articles written in the English language were obtained. We analyzed the data from the available literature.

### 2.3. Data Collection

We assessed all the included studies. Each study was reviewed, and then the main characteristics of the studies were recorded. We have summarized the study findings for serological, MRI, genetic, and viral examinations in Appendix A, respectively.

### 2.4. Statistical Analysis

Statistical analysis was performed using SPSS ver.26 (IBM, Armonk, NY, USA). We performed linear regression analysis to elucidate the hearing prognosis of patients who developed sensorineural hearing loss associated with COVID-19.

## 3. Results

### 3.1. Serological Examinations

We identified 61 studies and selected nine for this study (Appendix A). White blood cell counts and their subtypes, such as neutrophils, are known as classical inflammatory markers in patients with cardiovascular diseases [13]. In particular, neutrophil-to-lymphocyte ratio (NLR) is a new and quick inflammatory marker and has been introduced as a potential marker to determine inflammation and prognosis in patients with cardiac disorders [14,15]. The NLR was evaluated to investigate the relationship between SSNHL and inflammation. The mean NLR in patients with SSNHL was significantly higher than that in age- and sex-matched healthy individuals with normal hearing [16,17]. When patients with SSNHL were divided into two groups: recovered and unrecovered, according to their response to corticosteroid therapy, the mean NLR in the unrecovered patients was significantly higher than that in the recovered patients [16]. Other inflammatory markers, such as calprotectin, were evaluated in patients with SSNHL. Calprotectin, released from neutrophils, has been used in many studies as a biomarker for indicating the presence of inflammation [18,19,20]. The serum calprotectin value in patients with SSNHL was significantly higher than that in a healthy control group with normal hearing. When patients with SSNHL were classified into subgroups according to the severity of hearing loss, calprotectin values in the severe group were significantly higher than those in the mild and moderate groups. Serum calprotectin value was also associated with prognosis [21]. Serum calprotectin levels in patients with recovered hearing or partially recovered hearing significantly decreased after treatment compared with patients with unrecovered hearing [21]. Thus, NLR levels and serum calprotectin values were related to the prognosis of SSNHL, indicating that the mechanism underlying SSNHL is associated with inflammation. The levels of inflammatory cytokines, such as tumor necrosis factor-α (TNF-α), and the expression of dendritic cell markers, such as a cluster of differentiation 86 (CD86), were evaluated in patients with SSNHL. The mean TNF-α level in patients with SSNHL was significantly higher than that in control individuals [22,23]. CD86 expression in peripheral blood mononuclear cells (PBMCs) was significantly higher in patients with SSNHL than in control groups [23]. Circulating monocytes can be activated by proinflammatory cytokines, such as TNF-α, leading to their migration into tissues, and differentiation into a range of tissue dendritic cells or macrophages [24]. Thus, TNF-α may induce inflammation that enhances the differentiation of monocytes into dendritic cells probably in the cochlear tissue of patients with SSNHL.

The other markers associated with thrombosis were also evaluated in patients with SSNHL. The platelet-to-lymphocyte ratio (PLR) closely correlates with peripheral arterial diseases, such as arterial thrombosis and arteriosclerosis [25]. The mean PLR in patients with SSNHL was significantly higher than that in control individuals. After treatment with a corticosteroid, the mean PLR in recovered patients significantly decreased compared to that in unrecovered patients [17,26]. The elevated serum lipid levels increased the risk of arterial thrombosis [27]. Among the lipids, low-density lipoprotein cholesterol (LDL-C) is correlated with the development of atherosclerotic cardiovascular diseases [28], while high-density lipoprotein cholesterol (HDL-C) has atheroprotective effects [29]. Thus, the ratio of LDL-C and HDL-C was one of the predictors of atherosclerosis. The LDL-C/HDL-C ratio was evaluated in patients with SSNHL to investigate the correlation between the ratio and the degree of hearing recovery. When patients with SSNHL were classified into four groups: complete, partial, slight, and no recovery after treatment, the LDL-C/HDL-C ratio was associated with recovery outcome showing an upward trend from complete recovery to slight recovery group with statistically significant difference [30,31]. Thus, PLR levels and LDL-C/HDL-C ratio were correlated with the prognosis of SSNHL, indicating that the mechanism underlying SSNHL is associated with thrombosis and can lead to vascular obstruction probably in the cochlea. In addition, LDL-C correlates with oxidative stress, while HDL-C has antioxidant effects [32,33]. Thus, the LDL-C/HDL-C ratio is also associated with inflammation, supporting the above-mentioned hypothesis that inflammation is a possible mechanism underlying SSNHL.

Moreover, the role of endoplasmic reticulum (ER) stress was evaluated to investigate the pathogenesis of SSNHL at the molecular level. ER stress was characterized by accumulation of aberrant proteins in the ER when cells were exposed to stressors. The stressed cells respond to ER stress by inducing unfolded protein response (UPR). UPR helps increase the ability of ER to fold proteins properly, regulate protein translation, and induce apoptotic cell death if cells cannot be restored [34]. UPR mainly comprises three distinct signal transduction pathways mediated through protein kinase RNA-like ER kinase (PERK), inositol-requiring protein-1, and activating transcription factor-6 (ATF6). Of these, the pathway mediated through PERK is predominant. The increased presence of aberrant proteins in the ER activates PERK through its phosphorylation, resulting in the phosphorylation of eukaryotic initiation factor 2a (eIF2a), which increases the expression of activating transcription factor 4 (ATF4) and C/EBP homologous protein (CHOP) [35]. Protein levels of phosphorylated PERK (p-PERK), phosphorylated eIF2a (p-eIF2a), ATF4, and CHOP were investigated in PBMCs extracted from patients with SSNHL using Western blotting. Levels of p-PERK, p-eIF2a, ATF4, and CHOP were significantly elevated in PBMCs extracted from patients with SSNHL compared with those from healthy controls. After treatment with a corticosteroid, levels of ATF4 and CHOP were significantly reduced in PBMCs from improved patients, although neither ATF4 nor CHOP changed significantly in PBMCs from unimproved patients [36]. Thus, UPR mediated through PERK may play a significant role in the pathogenesis of SSNHL, and the response to ER stress may be one of the factors affecting prognosis after corticosteroid treatment.

### 3.2. MRI Examinations

We identified 69 studies and selected three for this study (Appendix A). The capillary network in the cochlea has a semi-permeable barrier called the blood-labyrinth barrier. The barriers are composed of endothelial cells with tight junctions and tightly regulate ion composition in the endolymph and perilymph [37]. The ion concentrations must be maintained for mechanosensory transduction of hair cells with maximum sensitivity [38]. Several studies have shown that the perilymphatic space of the cochlea is enhanced on MRI after intravenous administration of gadolinium-based contrast agents when the integrity of the barrier is disrupted [39]. Thus, disruption of the blood labyrinth barrier was examined using MRI to investigate the pathogenesis of sensorineural hearing loss. For instance, postcontrast fluid-attenuated inversion-recovery (FLAIR)-MRI examination revealed that patients with genetic hearing loss caused by NLRP3 mutations showed cochlear enhancement, suggesting that their hearing loss was caused by the disruption of the blood-labyrinth barrier via cochlear inflammation [40]. MRI with strong magnetic fields and new sequences also sheds light on the pathogenesis of SSNHL. Some patients with SSNHL showed cochlear enhancement on postcontrast FLAIR-MRI. Furthermore, some patients showed a high intensity signal in the cochlea without contrast in T1-weighted MR images or on FLAIR-MRI sequences [41,42], while healthy controls with normal hearing showed no signal alteration in T1-weighted or FLAIR-MRI sequences, even after the administration of gadolinium. Thus, patients with SSNHL showed several features on MR images. According to the analysis of these images on T1-weighted and FLAIR sequences, the images can be classified into the following three patterns: Pattern 1, the presence of high intensity-signal on T1-weighted and FLAIR sequences; Pattern 2, the presence of high intensity-signals on T1-weighted images, but no signal alteration on FLAIR sequences; and Pattern 3, the presence of enhancement after gadolinium injection that can be associated with either pattern 1 or pattern 2. Pattern 1 represents the presence of intracellular and extracellular methemoglobin, a dysfunctional form of hemoglobin, potentially consistent with intracochlear hemorrhage [42]. Pattern 2 represents increased protein exudate in the membranous fluid, potentially consistent with an acute inflammatory process. Pattern 3 is considered a result of the disruption of blood-labyrinth barrier [42].

Other studies evaluated cerebral white matter hyperintensity (WMH) on T2-weighted MR images. WMH reflects silent brain infarcts and white matter lesions, and are common in healthy older adults. The presence of WMH is believed to be related to cerebral small vessel diseases and reportedly increases the risk of recurrent stroke [43]. The relationship between the severity of WMH and the prognosis of SSNHL was investigated in patients with SSNHL. When patients with SSNHL were classified into three subgroups according to the severity of WMH, hearing gain of the mildest subgroup 1 was more prevalent than that in control groups characterized by the absence of WMH after corticosteroid therapy, while hearing gain in subgroups 2 and 3 was not significantly different from that in the control group [44]. Thus, the presence of mild WMH was associated with better treatment response and good prognosis [44]. Since WMH was mainly caused by local hemodynamic alteration, the presence of mild WMH may mean that hemodynamic change is likely to be reversible even in the cochlea. The relationship between the severity of WMH and the prognosis of SSNHL is not necessarily direct; however, these data indicate that vascular hemodynamics change is one of the factors involved in the pathogenesis of SSNHL. A meta-analysis investigating the effect of SSNHL on the incidence of stroke revealed that the presence of SSNHL is a stronger risk factor for stroke compared with the presence of other sensorineural hearing loss, including age-related hearing loss [45]. This is also consistent with the speculation that vascular hemodynamics change is associated with SSNHL.

### 3.3. Genetic Examinations

We identified 36 studies and selected 14 for this study (Appendix A). Several studies have revealed that genetic polymorphisms are associated with susceptibility to SSNHL. Most studied polymorphisms are involved in genes related to thrombosis, inflammatory response, and oxidative stress [46,47] (Table 1). Thrombosis-related genes include coagulation-related genes (F2, F5), platelet-related genes (ITGA2, ITGB3), and a homocysteine metabolism-related gene (MTHFR). F2 and F5 genes encode coagulation factor 2 (also known as prothrombin) and coagulation factor 5, respectively. Both of them are key enzymes involved in the coagulation process. Single-nucleotide polymorphisms (SNPs) in F2 (rs1799963) and F5 (rs6025) were associated with susceptibility to SSNHL [46,48]. ITGA2 and ITGB3 encode integrin beta 2 and 3 proteins, respectively. Integrins are heterodimeric transmembrane receptors comprising alpha- and beta-subunits that facilitate cell–cell adhesion. SNPs in ITGA2 and ITGB3 increase the risk of arterial thrombosis by affecting platelet receptors [49]. SNPs in ITGA2 (rs1126643) and ITGB3 (rs5918) were associated with susceptibility to SSNHL [8,46]. MTHFR encodes an enzyme called methylenetetrahydrofolate reductase, which plays an important role in the metabolic process that converts homocysteine to methionine. High levels of homocysteine can damage vascular endothelial cells, leading to thrombosis. Thus, high homocysteine levels are a risk factor for cardiovascular diseases [50]. An SNP in MTHFR (rs1801133, rs1801131) was significantly correlated with susceptibility to SSNHL [51].

Inflammatory response-related genes include a tumor-necrosis-related gene (LTA) and interleukin-related genes (IL1A, IL1B, IL4R, IL6). LTA encodes lymphotoxin-alpha (also known as tumor necrosis factor-beta), which plays an important role in immune regulation and inflammatory response. IL1A, IL1B, IL4R, and IL6 encode interleukin-1 alpha, interleukin-1 beta, interleukin-4 receptor, and interleukin-6, respectively. Interleukins are cytokines expressed or secreted by leukocytes, such as lymphocytes, monocytes, and macrophages, and play key roles in immunity and inflammatory response. SNPs in LTA (rs909253), IL1A (rs1800587), IL1B (rs16944, rs1143634), IL4R (rs180275), and IL6 (rs1800796, rs1800795) are correlated with increased risk of SSNHL [52,53,54,55,56,57].

Oxidative stress-associated genes include the superoxide dismutase gene (SOD1), glutathione-related genes (GPX3), and iron homeostasis gene (SLC40A1). SOD1 encodes superoxide dismutase 1, which is one of the three superoxide dismutases responsible for inactivating superoxide radicals. GPX3 and GSTP1 encode glutathione peroxidase 3 and glutathione S-transferase pi 1. Glutathione is a tripeptide composed of glutamic acid, cysteine, and glycine, and has the function of preventing damages to cellular components caused by reactive oxygen species (ROS), including superoxide anion, hydrogen peroxide, and hydroxyl radicals. Glutathione exists in two forms: reduced glutathione (GSH) and oxidized glutathione (GSSH). GSH can convert toxic peroxide to non-toxic hydroxy compounds through its conversion to GSSH. This conversion is catalyzed by glutathione peroxidase. Thus, glutathione peroxidase has the function of oxygen radical-metabolizing enzymes [58]. SNPs in SOD1 (rs4998557), and GPX3 (rs3805435) are significantly different between patients with SSNHL and healthy control individuals [59,60]. SLC40A1 encodes ferroportin-1 protein. Ferroportin-1 is a transmembrane protein that exports nonheme iron from the inside of a cell, and is associated with iron homeostasis. The excess iron is associated with increased ROS by enhancing redox cycling and free radical formation [61]. An SNP in SLC40A1 (rs11568351) was associated with a significantly increased risk of developing SSNHL [62].

Besides evaluating the correlation between genetic SNPs and risk of developing SSNHL, the expression levels of microRNAs (miRNAs) were examined in patients with SSNHL. The miRNAs are non-coding RNAs composed of 20–25 nucleotides, which play a significant role in gene expression and regulation. Dysregulation of miRNAs is reportedly correlated with congenital and age-related hearing loss [63]. Eight miRNAs, including hsa-miR-590-5p, hsa-miR-186-5p, hsa-miR-195-5p, hsa-miR-140-3p, hsa-miR-128-3p, hsa-miR-132-3p, hsa-miR-375-3p, and hsa-miR-30a-3, were significantly differentially expressed in patients with SSNHL compared to control subjects with normal hearing [64]. Most of the miRNAs were highly expressed in the nervous system, and their putative target messenger RNAs are abundant in cellular signaling pathways, such as phosphatidylinositol 3-kinase/protein kinase B (PI3K/Akt) and RAS/mitogen-activated protein kinase (MAPK) pathways [65,66,67]. These findings indicate that these cellular signaling pathways may be responsible for the disruption of auditory signal transduction in patients with SSNHL.

**Table 1 jcm-11-06387-t001:** Genetic polymorphisms associated with susceptibility to sudden sensorineural hearing loss.

Gene Classification	Gene Description	Gene	Polymorphism	Position (GRCh38.p13)	Alleles	Allele Frequency *	The Number of Individuals Studied	Reference
Thrombosis-related genes	Coagulation-related gene	*F2*	rs1799963	chr11:46739505	G > A	G = 0.989, A = 0.011	Patients = 100, Controls = 200	Capaccino P. et al. (2007) [46]
		*F5*	rs6025	chr1:169549811	C > T	C = 0.977, T = 0.023	Patients = 100, Controls = 200	Capaccino P. et al. (2007) [46]
							Patients = 56, Controls = 95	Görür K. et al. (2005) [48]
	Platelet-related genes	*ITGA2*	rs1126643	chr5:53051539	C > T	C = 0.611, T = 0.389	Patients = 118, Controls = 161	Ballesteros F. et al. (2012) [8]
							Patients = 85, Controls = 85	Rudack C. et al. (2004) [68]
		*ITGB3*	rs5918	chr17:47283364	T > C	T = 0.852, C = 0.148	Patients = 100, Controls = 200	Capaccino P. et al. (2007) [46]
	Homocysteine metabolism-related gene	*MTHFR*	rs1801133	chr1:11796321	G > A	G = 0.660, A = 0.340	Patients = 100, Controls = 200	Capaccino P. et al. (2007) [46]
							Patients = 67, Controls = 134	Rubino E. et al. (2005) [51]
			rs1801131	chr1:11794419	T > G	T = 0.696, G = 0.304	Patients = 100, Controls = 200	Capaccino P. et al. (2007) [46]
							Patients = 67, Controls = 134	Rubino E. et al. (2005) [51]
Inflammatory response-related genes	Inflammatory response-related genes	*LTA*	rs909253	chr6:31572536	A > G	A = 0.675, G = 0.325	Patients = 97, Controls = 587	Um J.-Y. et al. (2010) [52]
		*IL1A*	rs1800587	chr2:112785383	G > A	G = 0.715, A = 0.285	Patients = 72, Controls = 2202	Furuta T. et al. (2011) [53]
		*IL1B*	rs16944	chr2:112837290	A > G	A = 0.357, G = 0.643	Patients = 102, Controls = 595	Um J.-Y. et al. (2013) [54]
			rs1143634	chr2:112832813	G > A	G = 0.772, A = 0.228	Patients = 102, Controls = 595	Um J.-Y. et al. (2013) [54]
		*IL4R*	rs180275	chr7:93906158	G > A	G = 0.900, A = 0.100	Patients = 97, Controls = 613	Nam S.I. et al. (2006) [55]
		*IL6*	rs1800796	chr7:22726627	G > C	G = 0.929, C = 0.071	Patients = 75, Controls = 165	Tian G. et al. (2018) [56]
			rs1800795	chr7:22727026	C > G	C = 0.361, G = 0.639	Patients = 87, Controls = 107	Cadoni G. et al. (2015) [57]
Oxidative stress-associated genes	Superoxide dismutase gene	*SOD1*	rs4998557	chr21:31662579	G > A	G = 0.809, A = 0.191	Patients = 192, Controls = not written	Kitoh R. et al. (2016) [59]
	Glutathione-related genes	*GPX3*	rs3805435	chr5:151021735	T > C	T = 0.911, C = 0.089	Patients = 416, Controls = 225	Chien C.-Y. et al. (2017) [60]
	Iron homeostasis gene	*SLC40A1*	rs11568351	chr2:189580468	G > C	G = 0.801, C = 0.199	Patients = 200, Controls = 400	Castiglione A. et al. (2015) [62]

* Allele frequency was searched on db SNP database (https://www.ncbi.nlm.nih.gov/snp/ (accessed on 16 October 2022)).

### 3.4. Viral Examinations

We identified 86 studies and selected 23 for this study (Appendix A). Sensorineural hearing loss occurs during viral infections, including herpes simplex virus (HSV), mumps virus, measles virus, rubella virus, hepatitis virus, human immunodeficiency virus, Lassa virus, and enterovirus [69,70,71,72]. When virus-induced sensorineural hearing loss occurs in asymptomatic individuals with sudden onset, the sensorineural hearing loss can be diagnosed as SSNHL. Thus, viral infection is speculated as one of the etiologies of SSNHL. To date, several studies have proposed a possible association between viral infection and SSNHL [69,70]. Three hypotheses have been proposed to explain how viral infection causes sensorineural hearing loss: the first is through direct viral invasion of the cochlear nerve or cochlear tissue; the second is through the reactivation of the latently virus-infected cochlear tissue; the third is through an adaptive immune response triggered by antigens in the cochlea that were produced by systemic or distant viral infection [73]. However, studies on the sensorineural hearing loss caused by viral infection are limited, and most of them are case reports or case series. Recently, sensorineural hearing loss associated with COVID-19 has been the center of attention, and there have been multiple reports. We summarized SSNHL caused by COVID-19 to reveal its clinical characteristics and investigate the pathogenesis of sensorineural hearing loss caused by SARS-CoV-2 infection.

After the first report of COVID-19 cases in China, we retrieved articles comprising 18 patients who developed sensorineural hearing loss associated with COVID-19 (searched on January 19, 2022) [74,75,76,77,78,79,80,81,82,83,84,85,86,87]. Eighteen patients (10 men and 8 women) with a mean age of 45 years (18–68 years), were included (Table 2). Hearing loss was unilateral in 13 patients (72.2%) and bilateral in five patients (28.8%), and accompanied by symptoms such as tinnitus, dizziness, and facial palsy. The degree of sensorineural hearing loss ranged from mild to severe, showing various patterns of audiograms, such as flat-type, low-frequency type, and high-frequency type. Their hearing thresholds were confirmed by auditory brainstem response in nine patients. Distortion product otoacoustic emissions were absent in all nine patients examined, suggesting that sensorineural hearing loss is of the cochlear origin. MRI was performed in 11 patients, of which four patients demonstrated the following (Table 3): A 60-year-old man showed increased contrast in the right cochlea on T1 post-contrast sequence and partially decreased fluid signal in the basal turn of the right cochlea on T2 sequence, suggesting cochlear inflammation [74]: A 62-year-old woman exhibiting hearing loss and facial palsy showed enhancement of left seventh and eighth nerves on T1 post-contrast sequence, indicating inflammation of the nerves [80]: An 18-year-old woman presented high intensity in the left cochlea, vestibule, lateral, and superior semicircular canals on both FLAIR and T1 sequences without enhancement on T1 post-contrast sequence, indicating hemorrhagic lesion in these tissues [82]: An 84-year-old man demonstrated high intensity in the right cochlea, vestibule, and semicircular canals on both FLAIR and diffusion-weighted sequences with normal intensity on T1 sequences, indicating inflammation of these tissues [85]. Thus, MRI findings suggest that sensorineural hearing loss associated with COVID-19 is mainly of the cochlear origin through inflammation. We analyzed hearing prognosis in 14 ears from 12 patients for whom hearing thresholds could be evaluated before and after corticosteroid therapy. The prognosis was classified according to Siegel’s classification: Class-I (complete recovery), final averaged hearing threshold ≤ 25 dB regardless of hearing gain; Class-II (partial recovery), final averaged hearing threshold between 25 dB and 45 dB with hearing gain ≥ 15 dB; Class III (slight recovery), final averaged hearing threshold > 45 dB with hearing gain ≥ 15 dB; Class IV (no improvement), final averaged hearing threshold > 25 dB with hearing gain < 15 dB [88]. We defined Class-I as cured, Class-II and Class-III as improved, and Class-IV as unchanged; based on this criterion, five ears were cured, six ears were improved, and three ears were unchanged (Table 3). Their average pre-treatment hearing thresholds were associated with their hearing prognoses, which showed upward trend from Class-I to Class-IV group with a statistically significant difference (Figure 1, linear regression analysis, R = 0.88, *p* < 0.001).

Mustafa evaluated the hearing threshold in 20 asymptomatic patients diagnosed with COVID-19. The infected patients showed predominantly elevated hearing thresholds at 4, 6, and 8 kHz in pure-tone audiometry compared with 20 non-infected healthy controls. The amplitude of transiently evoked otoacoustic emission was also predominantly decreased in the infected group [89]. These findings indicated that asymptomatic patients might have shown sensorineural hearing loss due to cochlear damage and would have been diagnosed with SSNHL. In addition, SSNHL was reported in three patients as an otolaryngologic adverse event of COVID-19 vaccination. Although the etiology of SSNHL was unknown, the occurrence of SSNHL within three days after COVID-19 vaccination suggests that vaccination was the main cause of SSNHL [90]. Nikolaos et al. reported a case of right SSNHL that started 2 days after a second dose of Oxford-AstraZeneca COVID-19 vaccine. Despite no findings of thrombosis on head MRI or MR angiography, acetylsalicylic acid was administered in addition to glucocorticoids. Since the patient’s hearing was almost at full recovery 15 days after deafness, the authors suggested that acetylsalicylic acid might be considered for vaccine-induced sensorineural hearing loss [91]. Jeong et al. examined the mechanism underlying sensorineural hearing loss. They revealed that adult human inner ear tissues express the angiotensin-converting enzyme 2 (ACE2) receptor that is a target for the SARS-CoV-2 virus, and the transmembrane protease serine 2 (TMPRSS2) and FURIN cofactors required for virus entry. They established three human-induced pluripotent stem cells-derived in vitro models of the inner ear for infection: two-dimensional otic prosensory cells (OPCs) and Schwann cell precursors (SCPs), and three-dimensional inner ear organoids. Both OPCs and SCPs expressed ACE2, TMPRSS2, and FURIN, and were permissive to SARS-CoV-2 infection. Inner ear organoids expressed ACE2 [92]. These results indicate that SARS-CoV-2 can infect inner ear tissues and directly damage them.

Given the many reports that COVID-19 is associated with SSNHL, it is of interest to determine whether COVID-19 increases the number of SSNHL patients. Ling et al. in China reported an increase in SSNHL visits during the COVID-19 pandemic period from February to April 2020 compared to the same period in the previous three years [93]. Similarly, Italy and Turkey reported an increase in SSNHL during the COVID-19 pandemic [94,95]. In contrast, some studies have found no increase in SSNHL incidence due to COVID-19 [96,97], while another has reported a decrease [98]. Thus, it is still unclear whether SARS-Cov-2 contributes to the incidence of SSNHL.

## 4. Discussion

SSNHL is routinely encountered and is one of the most common emergent diseases in otolaryngology clinics. The etiology of SSNHL remains unknown; however, findings from serological, MRI, genetic, and viral examinations can suggest possible underlying mechanisms causing SSNHL.

The first possible mechanism is thrombosis and resulting vascular obstruction in the cochlea. The markers associated with thrombosis, including PLR and LDL-C/HDL-C ratio, were correlated with the prognosis of SSNHL [17,26,30,31]. SNPs in thrombosis-related genes, including F2, F5, ITGB3, ITGB2, and MTHFR, were associated with susceptibility to SSNHL [8,46,48,50,68]. In addition, the presence of mild WMH was associated with the prognosis of SSNHL, and the presence of SSNHL is a strong risk factor for stroke [44]. Conversely, several studies have revealed that the presence of stroke is a significant risk factor for SSNHL [99,100], suggesting the hypothesis that SSNHL is caused by thrombosis and resulting vascular obstruction in the cochlea. Migraine-associated SSNHL is thought to be caused by vasospasm of cochlear vasculature or increased vascular permeability in the cochlea [101]. A report of the long-term course of migraine-associated SSNHL in 21 patients reported a mean age of 64 ± 11 years [101]. Large cohort studies in Taiwan and Korea have also reported that migraine is a risk factor for SSNHL in all age groups [102,103]. These findings may support the hypothesis that thrombus formation in the cochlea causes hearing loss. Levels of PERK pathway-related proteins, including p-PERK, p-eIF2a, ATF4, and CHOP, were significantly elevated in PBMCs extracted from patients with SSNHL compared with those from healthy controls [36]. Thus, UPR mediated through PERK pathway may play a significant role in the pathogenesis of SSNHL. To date, several studies have suggested that ER stress and UPR induced by high blood cholesterol levels are associated with the pathogenesis of atherosclerotic lesions [104,105]. Some patients with SSNHL may have high blood cholesterol levels represented by a high LDL-C/HDL-C ratio that induces ER stress and UPR, resulting in atherosclerotic lesions and thrombosis of blood vessels in the cochlea. The oxidative stress in the red blood cells (RBCs) is reportedly associated with thrombosis. Elevated ROS in RBCs affects the membrane structure of RBCs, causing a loss of membrane integrity. These changes impair RBC function in thrombosis, leading to a hypercoagulable state through enhanced RBC coagulation, thereby enhancing binding to vascular endothelial cells [106]. Thus, SNPs in oxidative stress-associated genes, including SOD1, GPX3, GSTP1, and SLC40A1, are associated with susceptibility to SSNHL [60,61,63], possibly because of conversion of RBCs into a hypercoagulable state.

The second possible mechanism is an asymptomatic viral infection and resulting damage to the cochlea. When virus-induced sensorineural hearing loss occurs in asymptomatic individuals with sudden onset, the sensorineural hearing loss can be diagnosed as SSNHL without other organ manifestations. This hypothesis is supported by the findings that asymptomatic patients diagnosed with COVID-19 showed predominantly elevated hearing thresholds at high frequencies in pure-tone audiometry [89]. The human-induced pluripotent stem cells-derived in vitro models of the inner ear revealed that inner ear tissues express at least one of the markers among ACE2, TMPRSS2, and FURIN, and are permissive to SARS-CoV-2 infection [92]. Mechanisms underlying virus-induced hearing loss remain unclear; however, some studies have investigated the pathogenesis underlying sensorineural hearing loss associated with COVID-19. MR images of patients with SARS-CoV-2-induced sensorineural hearing loss revealed the presence of inflammation in the inner ear [74,80,85]. The inflammation would have been induced through direct viral invasion, rather than through reactivation of the latent virus or through an adaptive immune response triggered by antigens in the inner ear that were produced by a systemic or distant viral infection, because the averaged latent period of SARS-CoV-2-induced sensorineural hearing loss was approximately 8 days. The direct viral invasion of the inner ear was also documented in the HSV labyrinthitis mouse model. The mouse model was established through the inoculation of HSV type 1 or 2 into the middle ear, and the mouse model showed hearing loss and vestibular dysfunction. The histopathological examination of the inner ear revealed that the epithelial cells in the stria vascularis and the vestibular ganglions were infected, and cells in the organ of Corti were apoptotic [107].

The third possible mechanism is cochlear inflammation and the resulting damage to the cochlea. The markers associated with inflammation, including NLR and calprotectin, were related to the prognosis of SSNHL [16,17,20]. Levels of TNF-α and CD86 expression in PBMCs were significantly higher in patients with SSNHL than in control groups [22,23]. SNPs in inflammation-related genes, including LTA, IL1A, IL1B, IL4R, and IL6, were correlated with an increased risk of SSNHL [52,53,54,55,56,57]. MR images represented by the presence of high intensity signals on T1-weighted images and no signal alteration on FLAIR sequences were observed in some patients with SSNHL, indicating acute cochlear inflammation. The presence of enhancement on post-contrast sequence as observed in some patients with SSNHL suggested a disruption of the blood-labyrinth barrier [41]. This disruption can be seen in a wide range of conditions, including genetic defects, inflammation, loud sound trauma, and aging [108]; however, it can be caused by inflammation. The unaddressed question of the mechanism is whether cochlear inflammation is primarily induced through immune system dysfunction, such as aberrant activation of innate immune responses or autoimmune response, or secondarily induced through viral infection or thrombosis [109,110,111]. Patients with SSNHL caused by primary inflammation of the cochlea may be rare because sensorineural hearing loss caused by aberrant activation of innate immune responses or autoimmune response has been reported in only a few individuals [110,112].

Thus, the mechanism underlying SSNHL may be associated with thrombosis, viral infection, or inflammation. However, the fact that available studies in this review used small sample sizes and there were contrary data regarding serological, MRI, genetic, and viral examinations suggest that additional research is needed to elucidate the etiology of SSNHL.

## 5. Conclusions

We updated the findings obtained from serological, MRI, genetic, and viral examinations. We proposed the following underlying mechanisms of SSNHL: thrombosis and resulting vascular obstruction in the cochlea; asymptomatic viral infection and resulting damage to the cochlea; and cochlear inflammation and resulting damage to the cochlea. Thrombosis and viral infection are predominant, and cochlear inflammation can be secondarily induced through viral infection or even thrombosis. Future investigations elucidating the etiology of SSNHL are necessary to improve the prognosis of patients with SSNHL.

## Figures and Tables

**Figure 1 jcm-11-06387-f001:**
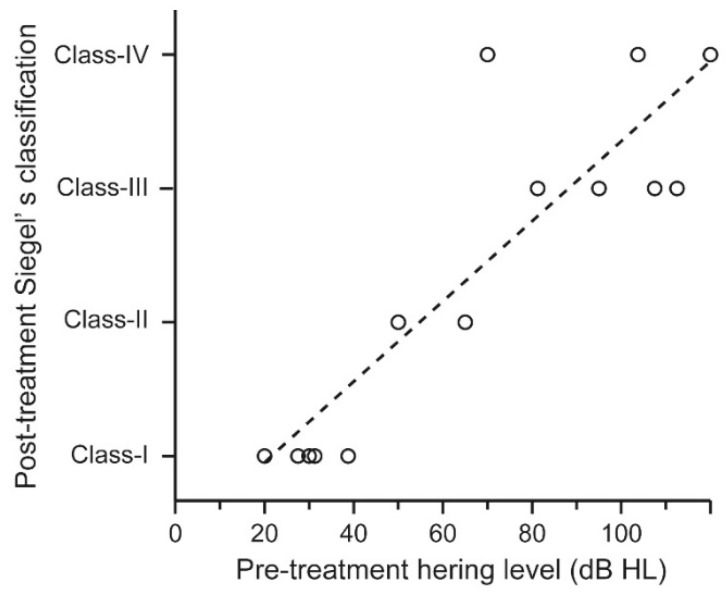
Hearing prognosis of patients who developed sensorineural hearing loss associated with COVID-19. The hearing prognoses of 18 patients who developed sensorineural hearing loss associated with COVID-19 were plotted according to their average pre-treatment hearing thresholds. Their pre-treatment hearing levels were associated with their hearing prognoses, which showed upward trend from Class-I to Class-IV groups with statistically significant difference (linear regression analysis, R = 0.88, *p* < 0.001).

**Table 2 jcm-11-06387-t002:** Patient demographics in coronavirus disease-related sudden sensorineural hearing loss.

Mean Age (Years)		45 (Range: 18–68)
Sex (number, percentage)	Men Women	10 (55.6%) 8 (44.4%)
Affected side (number, percentage)	Unilateral Bilateral	13 (72.2%) 5 (28.8%)
Other symptoms (number, percentage)	Tinnitus Vertigo or dizziness Facial palsy	11 (61.1%) 4 (22.2%) 2 (11.1%)

**Table 3 jcm-11-06387-t003:** Characteristics of coronavirus disease-related sudden sensorineural hearing loss.

Latent Period of Sensorineural Hearing Loss	Approximately 8 Days
Magnetic resonance imaging findings	Increased contrast in the right cochlea on T1 post-contrast sequence and partially decreased fluid signal in the basal turn of the right cochlea on T2 sequence Enhancement of the left seventh and eighth nerves on T1 post-contrast sequence High intensity in the left cochlea, vestibule, lateral, and superior semicircular canals on both fluid-attenuated inversion-recovery (FLAIR) and T1 sequences without enhancement on T1 post-contrast sequence High intensity in the right cochlea, vestibule, and semicircular canals on both FLAIR and diffusion-weighted sequences with normal intensity on T1 sequences
Treatment	Systematic or intratympanic administration of corticosteroids, hyperbaric oxygen therapy
Prognosis	Cured, 36%; improved, 43%; unchanged, 21%

## Data Availability

Not applicable.

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
