# Peer review of "Update on Findings about Sudden Sensorineural Hearing Loss and Insight into Its Pathogenesis"

_jcm, 2022, doi:10.3390/jcm11216387_

Round 1

Reviewer 1 Report

Since the authors considered only serological, MRI, genetic, and viral etiology, readers could overlook other causes of SSNHL, such as acute noise exposure. The known causes of SSNHL that are not reviewed by the authors include autoimmune disease, noise exposure, physical damage, and acoustic tumor. 

As mentioned by the authors in the manuscript, since it is still impossible to assess the exact etiologies of SSNHL, , a further review papers that considers all indirectly revealed etiologies of SSNHL is needed.

Author Response

Our response:

Thank you for your suggestion.

In this article, we focused on SSNHL to explore the unknown etiology of Idiopathic sudden sensorineural hearing loss (ISSNHL). As you point out, the known causes of SSNHL, including autoimmune disease, noise exposure, physical damage, and acoustic tumor, are not included in this article. We have revised the text to explain this point clearly and avoid any confusion.

Lines 44~46: Despite extensive investigation, the etiology of SSNHL cannot be determined in approximately 90% of patients and is referred to as Idiopathic sudden sensorineural hearing loss (ISSNHL).

Lines 59~61: The purpose of this review was to understand the etiology of SSNHL and elucidate the unknown etiology of ISSNHL. For this reason, SSNHL caused by the autoimmune disease, noise exposure, physical damage, and acoustic tumor were not included.

Reviewer 2 Report

An overall great effort, but some modifications should be made. in line 377 authors discuss the relationship of COVID-19 and SSNHL, but they say nothing about the possible connection of vaccination against COVID-19 and SSNHL.''Tsetsos N, Poutoglidis A, Vlachtsis K, Kilmpasanis A, Gougousis S. Sudden Sensorineural Hearing Loss Following the Second Dose of COVID-19 Vaccine. Cureus. 2021 Aug 25;13(8):e17435.'' This high cited article discuss a case and makes a literature review. Authors should make an addition here and cite the article to strengthen the evidence.

Author Response

Our response:

Thank you for your suggestion. We added the suggested article by Tsetsos N et al.

Lines 351~357: Nikolaos et al. reported a case of right SSNHL that started 2 days after a second dose of Oxford-AstraZeneca COVID-19 vaccine. Despite no findings of thrombosis on head MRI or MR angiography, acetylsalicylic acid was administered in addition to glucocorticoids. Since the patient's hearing was almost at full recovery 15 days after deafness, the authors suggested that acetylsalicylic acid might be considered for vaccine-induced sensorineural hearing loss [91].

Reviewer 3 Report

The authors updated the latest evidence for the pathogenesis of SSNHL from serological, MRI, genetic, and viral examinations. This is a well written and excellent review. Several concerns should be addressed

A reference should be cited for line 169~174.

A reference should be cited for line 184~188.

A reference should be cited for line 89~94.

Line 421~422, these sentences could be summarized. Avoid wordy and redundant phrases

Line 425: prognosis of patients with SSNHL.

Author Response

A reference should be cited for line 169~174.

A reference should be cited for line 184~188.

A reference should be cited for line 89~94.

Our response:

Thank you for your suggestions. We have added suitable citations.

Lines 106~112: When patients with SSNHL were classified into subgroups according to the severity of hearing loss, calprotectin values in the severe group were significantly higher than those in the mild and moderate groups. Serum calprotectin value was also associated with prognosis [21]. Serum calprotectin levels in patients with recovered hearing or partially recovered hearing significantly decreased after treatment compared with patients with unrecovered hearing [21].

Lines 186~195: According to the analysis of these images on T1-weighted and FLAIR sequences, the images can be classified into the following three patterns: Pattern 1, the presence of high intensity-signal on T1-weighted and FLAIR sequences; Pattern 2, the presence of high intensity-signals on T1-weighted images, but no signal alteration on FLAIR sequences; and Pattern 3, the presence of enhancement after gadolinium injection that can be associated with either pattern 1 or pattern 2. Pattern 1 represents the presence of intracellular and extracellular methemoglobin, a dysfunctional form of hemoglobin, potentially consistent with intracochlear hemorrhage [42]. Pattern 2 represents in-creased protein exudate in the membranous fluid, potentially consistent with an acute inflammatory process. Pattern 3 is considered a result of the disruption of blood-labyrinth barrier [42].

Lines 199-206: The relationship between the severity of WMH and the prognosis of SSNHL was investigated in patients with SSNHL. When patients with SSNHL were classified into three subgroups according to the severity of WMH, hearing gain of the mildest subgroup 1 was more prevalent than that in control groups characterized by the absence of WMH after corticosteroid therapy, while hearing gain in subgroups 2 and 3 was not significantly different from that in the control group [44]. Thus, the presence of mild WMH was associated with better treatment response and good prognosis [44].

Line 421~422, these sentences could be summarized. Avoid wordy and redundant phrases

Line 425: prognosis of patients with SSNHL.

Our response:

Thank you for your suggestions. We have revised the wording to make it easier to understand.

Lines 458~461: We updated the findings obtained from serological, MRI, genetic, and viral examinations. We proposed the following underlying mechanisms of SSNHL: thrombosis and resulting vascular obstruction in the cochlea; asymptomatic viral infection and resulting damage to the cochlea; and cochlear inflammation and resulting damage to the cochlea.

Lines 463~465: Future investigations elucidating the etiology of SSNHL are necessary to improve the prognosis of patients with SSNHL.

Reviewer 4 Report

This is a good review of SSNHL with a scope on etiology. The review is not following a PRISMA guidelines and the criteria to select the studies and excluded studies are not mentioned. More details on the method section will be appreciated.

Methods section

Please, include a short description with the statistical analysis 

Results section

A suggest including the number of studies identified, and the final number of studies selected for the review.

The sentence “Due to the limited access to the cochlea and the difficulty in obtaining tissue samples, possible pathogenetic mechanisms of SSNHL have been investigated using peripheral blood” could be deleted. This idea is also mentioned in the abstract and the introduction.

The results are well structured in 4 sections: serological, MRI, genetic and viral examinations.

I would suggest to revise Table 1 including the list of genes, variants, their minor allelic frequencies and the number of individuals studied and OR with 95% confidence intervals. This information will add more clarify and value to the paper.

Regarding viral infections it coud be interesting if the authors could compare the incidence of Covid-19 infections with the incidence of SSNHL in China to infere if the number of cases have increased after Covid-19.

Discussion

Please include some comments about migraine in vascular risk factors and the age of onset. It is probably different the age of onset of viral, thrombosis and genetic patients.

Author Response

This is a good review of SSNHL with a scope on etiology. The review is not following a PRISMA guidelines and the criteria to select the studies and excluded studies are not mentioned. More details on the method section will be appreciated.

Our response:

Thank you for your suggestion. This study is a narrative review, not a systematic review. As such, it is not fully compliant with the PRISMA guidelines. However, many of the PRISMA guidelines references were used in the inclusion and exclusion criteria for the studies selected; thus, we have changed the description to a more detailed one based on this point.

Lines 63~85:

2.1. Database search

We obtained relevant literature published between 2013 and 2022 from PubMed and Embase using the following medical terms for the review article: hearing loss, SSNHL, neutrophil, monocyte, platelet, lipid for the sections of serological examinations; hearing loss, SSNHL, MRI for the section of MRI examinations; hearing loss, SSNHL, polymorphism, mutation, microRNA for the section of genetic examinations; and hearing loss, SSNHL, COVID-19, SARS-CoV-2, coronavirus, viral infection for the section of viral examinations. These words were combined with "and/or" for each section and searched. The retrieval scheme was based on a combination of medical subject headings (MeSH) terms and free words. Two reviewers (S.Y. and H.N.) independently assessed the eligibility of the studies and extracted the data.

2.2. Inclusion and exclusion criteria

We included articles from clinical series and reviews published in relevant journals with consistent and well-defined materials and methods. However, articles that were presented as abstracts, experts, letters, or comments with different themes of interest and repetitions were excluded. Articles not available in full-text or those that were not written in the English language were also excluded. Additional papers published before 2013 were identified through the references in the published literature. Full-text articles written in the English language were obtained. We analyzed the data from the available literature.

2.3. Data collection

We assessed all the included studies. Each study was reviewed, and then the main characteristics of the studies were recorded. We have summarized the study findings for serological, MRI, genetic, and viral examinations in Table S1, S2, S3, and S4, respectively.

Methods section

Please, include a short description with the statistical analysis.

Our response:

Thank you for your suggestion. We have added a short description on statistical analysis in the Methods section.

Lines 86~89:

2.4. Statistical analysis

Statistical analysis was performed using SPSS ver.26 (IBM, Armonk, NY, USA). We performed linear regression analysis to elucidate the hearing prognosis of patients who developed sensorineural hearing loss associated with COVID-19.

Results section

A suggest including the number of studies identified, and the final number of studies selected for the review.

Our response:

Thank you for your suggestions. We have added the number of studies identified and the final number of studies selected for this review.

Line 92: We identified 61 studies and selected nine for this study (Table S1).

Line 167: We identified 69 studies and selected three for this study (Table S2).

Line 217: We identified 36 studies and selected 14 for this study (Table S3).

Line 279: We identified 86 studies and selected 23 for this study (Table S4).

The sentence “Due to the limited access to the cochlea and the difficulty in obtaining tissue samples, possible pathogenetic mechanisms of SSNHL have been investigated using peripheral blood” could be deleted. This idea is also mentioned in the abstract and the introduction.

Our response:

Thank you for your suggestion. We have deleted the sentence.

I would suggest to revise Table 1 including the list of genes, variants, their minor allelic frequencies and the number of individuals studied and OR with 95% confidence intervals. This information will add more clarify and value to the paper.

Our response:

Thank you for your suggestion. We have added the list of genes, polymorphisms, positions, alleles, their minor allelic frequencies and the number of individuals studied.

Regarding viral infections it coud be interesting if the authors could compare the incidence of Covid-19 infections with the incidence of SSNHL in China to infere if the number of cases have increased after Covid-19.

Our response:

Thank you for your suggestion. There have been reports of an increase in cases of SSNHL during the COVID-19 pandemic, including a report from China. In contrast, some reports indicate that there was no increase in SSNHL cases during the COVID-19 pandemic, while another report indicates that there was a decrease. Thus, there is no consistent perspective yet.

Lines 367~374: Given the many reports that COVID-19 is associated with SSNHL, it is of interest to determine whether COVID-19 increases the number of SSNHL patients. Ling et al. in China reported an increase in SSNHL visits during the COVID-19 pandemic period from February to April 2020 compared to the same period in the previous three years [93]. Similarly, Italy and Turkey reported an increase in SSNHL during the COVID-19 pandemic [94,95]. In contrast, some studies have found no increase in SSNHL incidence due to COVID-19 [96,97], while another has reported a decrease [98]. Thus, it is still unclear whether SARS-Cov-2 contributes to the incidence of SSNHL.

Discussion

Please include some comments about migraine in vascular risk factors and the age of onset. It is probably different the age of onset of viral, thrombosis and genetic patients.

Our response:

Thank you for your suggestion. We have described the association between migraine and SSNHL. As you noted, age may vary depending on the cause of SSNHL. Despite multiple studies, it is still difficult to clearly distinguish the causes of SSNHL. For this reason, it is also difficult to compare ages, and we have decided not to include this information at this time because we believe it is likely to be an inaccurate description.

Lines 388~394: Migraine-associated SSNHL is thought to be caused by vasospasm of cochlear vasculature or increased vascular permeability in the cochlea [101]. A report of the long-term course of migraine-associated SSNHL in 21 patients reported a mean age of 64 ± 11 years [101]. Large cohort studies in Taiwan and Korea have also reported that migraine is a risk factor for SSNHL in all age groups [102,103]. These findings may support of the hypothesis that thrombus formation in the cochlea causes hearing loss.

Round 2

Reviewer 4 Report

Thank you for your revisions. I think that your paper has been improved and it  is suitable for publication.